# Mapping and Characterization of QTLs for Awn Morphology Using Crosses between “Double-Awn” Wheat 4045 and Awnless Wheat Zhiluowumai

**DOI:** 10.3390/plants10122588

**Published:** 2021-11-26

**Authors:** Tianxiang Liu, Xue Shi, Jun Wang, Jiawang Song, Enshi Xiao, Yong Wang, Xin Gao, Wenzhi Nan, Zhonghua Wang

**Affiliations:** 1Department of Agronomy, Northwest Agriculture and Forestry University, Xianyang 712100, China; ltxiang@nwafu.edu.cn (T.L.); shixue@nwafu.edu.cn (X.S.); bbrwangjun@nwafu.edu.cn (J.W.); xiaoenshixn@nwafu.edu.cn (E.X.); wangyong2114@nwafu.edu.cn (Y.W.); bestgaoxin@nwsuaf.edu.cn (X.G.); 2Key Laboratory of Plant Development and Stress Biology, Ministry of Education, School of Life Science, Shandong University, Qingdao 266237, China; songjiawang@sdu.edu.cn; 3College of Life Sciences, Yulin University, Yulin 719000, China

**Keywords:** awn, fine-mapping, zinc finger protein, wheat

## Abstract

Awns play important roles in seed dispersal, protection against predators, and photosynthesis. The characterization of genes related to the formation of awns helps understand the regulation mechanisms of awn development. In the present study, the “double-awn” wheat 4045, which features super-long lemma awns and long glume awns, and an awnless wheat line, Zhiluowumai, were used to investigate QTLs or genes involved in awn development. QTL analysis identified three loci—*Qawn-1D*, *Qawn-5A*, and *Qawn-7B*—using a population of 101 4045 × ZLWM F_2_ plants. Fine mapping with a total of 9018 progenies narrowed the mapping interval of *Qawn-5A* to an 809-kb region, which was consistent with the *B1* locus, containing five genes on chromosome 5AL. Gene structure and expression analysis indicated that *TraesCS5A02G542800* was the causal gene, which was subsequently verified by overexpression of *TraesCS5A02G542800* in a “double-awn” wheat, Yangmai20. The retained “double-awn” phenotype of transgenic plants suggested that *B1* represses the elongation but does not influence the emergence of the awns. Moreover, 4045 harbors a new allele of *B1* with a 261-bp insertion in the promoter region and a lack of the EAR2 motif in the encoding region, which influences several important agronomic traits. In this study, we identify two novel QTLs and a novel allele of *B1*, providing new resources for exploration of awn development.

## 1. Introduction

Awns are needle-like structures that emerge from the tips of the lemma in Poaceae grasses such as wheat (*Triticum aestivum* L.), rice (*Oryza sativa* L.), barley (*Hordeum vulgare* L.), and rye (*Secale cereale* L.). The barb-like structures on awns facilitate seed dispersal, which is considered the main function of the awns in wild grass species [1]. In wheat, the awns consist of sclerenchyma, two chlorenchyma zones, and three well-developed vascular bundles inside, with rows of stomata on the epidermis of the abaxial terminal [2]. The zonal distribution of the sclerenchyma tissue and three vascular bundles supports the solid awn structure [2], which helps to protect the seeds from being ingested by animals or birds [3]. Awns also play an important role as a photosynthetic organ because of their two chlorenchyma zones and large surface area [2,4,5]. By enhancing the carbon exchange rate and water use efficiency, awns facilitate photosynthesis and thus increase the grain filling efficiency [6,7,8]. Under extreme conditions, such as drought and heat stress, the awns contribute up to 16% of the total grain production [6,8,9,10].

Many genes involved in awn emergence and elongation have been characterized in grass species. In rice, *Awn-1* (*An-1*), a basic helix–loop–helix transcription factor that regulates cell division, promotes cell division of the lemma primordia and forms awn primordia [11]. *Awn-2* (*An-2*) encodes Lonely Guy Like protein 6 (OsLOGL6) and facilitates awn elongation by enhancing cell division but decreases grain production by reducing the grains per panicle and tillers per plant [12,13]. Furthermore, *Regulator of Awn Elongation 2* (*RAE2*), which encodes an Epidermal Patterning Factor-like protein (also known as *Grain Number, Grain Length, and Awn Development1*; *GAD1*), a YABBY transcription factor *DROOPING LEAF* (*DL*), and an auxin response factor *OsETT2*, has also been shown to be involved in awn formation or elongation [14,15,16]. The short awn 2 (*lks2*) gene encodes a SHI-family transcription factor. The awn length in a natural *lks2* variant of barley was reduced by about 50%, as compared to that of normal barley [17]. Duplication in the 4th intron of *Knox3* resulted in a hooked awn phenotype in barley [18].

In wheat, three predominant inhibitors of awn development—*B1* (*Tipped 1*), *B2* (*Tipped 2*), and *Hd* (*Hooded*)—have been identified and placed on chromosomes 5AL, 6BL, and 4BS, respectively [19,20,21,22]. Generally, the *B1* inhibitor, on its own, causes an awnletted phenotype for the top of the spikes; the *B2* allele reduces the length of awns on the top and bottom part of the spikes; the *Hd* allele reduces awn length consistently and causes a “hooked” awn phenotype [20]; while the combination of the three inhibitors results in an awnless phenotype [19]. Among the three inhibitors, *B1* is the most common one, inhibiting awn development in wheat cultivars and landraces [19,23,24,25]. Furthermore, a lack of the 3BS chromosome in Chinese Spring led to an awned phenotype, suggesting there exist unknown genes on 3BS that are involved in awn development [26]. In the wheat D genome donor *Aegilops tauschii* Coss., 5DS-located *Anathera* (*Antr*) also plays a dominant role in awn inhibition [27]. However, the molecular mechanisms of how the awn-development-involved genes regulate the formation of the awns remains largely unknown.

4045 is a wheat line that produces super-long awns that are longer than most of the hexaploid wheat cultivars. More surprisingly, the awns also extend from the tips of glumes, leading to the name glume-awns. The awn phenotype of 4045 is denoted “double-awn” in the following. Zhiluowumai (ZLWM) is an awnless hexaploid wheat line, derived from a wild blue-aleurone and awnless germplasm discovered in the Qingling mountains of China. Thus, the genome of ZLWM contains at least a part of genome sequences of wild wheat that are largely different from well-domesticated wheat cultivars. In this study, we crossed the “double-awn” wheat 4045 with the awnless wheat ZLWM to identify and characterize the genes involved in awn development in these two special germplasms. Genetic analysis showed that the awnless trait in the 4045 × ZLWM F_2_ population is incompletely dominant. QTL analysis identified three loci involved in awn development, of which the predominant QTL, *Qawn-5A*, is close to the *B1* locus. Fine mapping narrowed the mapping region of *Qawn-5A* to an 809-kb interval containing five annotated genes and confirmed that *Qawn-5A* is the same locus as *B1*. Transcriptional abundance and gene structure analysis strongly indicated that *TraesCS5A02G542800*, which encodes a C_2_H_2_ zinc finger protein with two EAR motifs, is the causal gene, which was further confirmed by overexpressing *TraesCS5A02G542800* in a “double-awn” wheat acceptor Yangmai20 (YM20). The transgenic plants retained the phenotype of “double-awn” with reduced awn lengths, indicating that *B1* suppresses the awn elongation but is not critical for the awn establishment. Moreover, we demonstrate that 4045 harbors a new haplotype of *TraesCS5A02G542800*, with a 261 bp insertion in the promoter region and a lack of the EAR2 motif in the encoding region, which is different from all of the detected wheat materials, providing a new resource for the investigation of awn development.

## 2. Results

### 2.1. Phenotypic and Genetic Analysis of Awn Trait in 4045 and ZLWM

To gain insight into the awn morphology of 4045 and ZLWM, the spikes at the flowering stage were observed and imaged (Figure 1A,B). The 4045 featured a “double-awn” phenotype with awns attached to both lemmas and glumes. The lemma awns were solid, curved, and super-long. In contrast, the glume awns were thinner, straight, and relatively short (Figure 1A,B). For regular awned hexaploid wheat cultivars, such as awned wheat cultivars Fielder and Aikang58 (AK58), the awn length varies from approximately 5–7 cm (see Appendix A). However, the longest glume awns of 4045 were about 5.5 cm, and its super-long lemma awns reached more than 10 cm (Figure 1A). ZLWM is known as a blue-aleurone wheat, of which the mature grains are colored blue (Figure 1C). The spike of ZLWM was much shorter, and only a few very short awn-like structures emerged from the lemma (Figure 1A). Furthermore, Fielder and AK58 presented short awn-like structures on the tip of glumes, while ZLWM showed no awns at all (Figure 1B). To further explore the genetic properties of the awn phenotype, we crossed 4045 with ZLWM. The 4045 × ZLWM F_1_ plants showed short top awns (Appendix A). Subsequently, the awn phenotype was investigated in a total of 806 F_2_ plants, among which 230 showed awned and 576 showed awnless phenotypes (Figure 1D). A chi-square test showed that χ^2^ = 5.375 > χ^2^_0.05,1_ = 3.841 (Figure 1C), indicating that the awn phenotype in 4045 × ZLWM was not genetically controlled by a single dominant gene. Unexpectedly, the glume awns and super-long awns always appeared simultaneously, and only six of the 806 F_2_ individuals showed this “double-awn” phenotype, implying that a complicated regulation network is involved in the determination and development of the super-long awn and glume awn phenotype.

### 2.2. Primary Mapping of the Awnless Trait

Previous studies have shown that *B1* plays a principal role in determining and regulating the development of awns [19,23,24,25]. Nevertheless, the awnless phenotype of ZLWM originated from a wild wheat germplasm. It is interesting to figure out whether the widespread *B1* awn inhibitor is also involved in awn development in ZLWM and 4045. Thus, 101 greenhouse-grown F_2_ individuals were sampled and genotyped by a wheat 90K SNP array [28] containing 81,587 SNPs. Removing the SNPs that were failed-genotyped in either of the parental lines, a total of 7636 SNPs showing polymorphisms were used for genetic map construction. The SNP markers were subsequently binned using the bin function of the Icimapping software to remove the redundant markers. As a result, 3413 bin markers were obtained after binning. The bin markers were then grouped by anchors, ordered using the nnTwoOpt strategy, and rippled with the default parameters of the Icimapping software. After, 126 unlinked markers were deleted and a high-density genetic map spanning 16439.19 cM was constructed using the 3287 bin markers from the 7636 SNPs (Table 1). In the genetic map, the D genome spanned 1938.55 cM (11.79%) and contained 334 bin markers from 925 SNPs, which was much shorter than the A (42.68%) and B (45.53%) genomes (Table 1). The lengths of each chromosome ranged from 53.65 cM (7D) to 1443.05 cM (3B). The number of bin markers on chromosomes ranged from 11 (7D) to 370 (5B); see Table 1.

Awn phenotypes of the 101 F_2_ individuals were investigated and used for mapping. As a result, three QTLs on 1D, 5A, and 7B, with the logarithm of odds (LOD) values of 3.27, 97.32, and 25.54, respectively, were identified (Table 2, Figure 2). This result confirmed the conclusion that the awn phenotype was not genetically controlled by a single dominant gene in 4045 × ZLWM (Figure 1D). Notably, *Qawn-5A* explained 91.22% of the phenotypic variation, whereas *Qawn-1D* and *Qawn-7B* explained only 0.63% and 2.76%, respectively (Table 2), indicating that *Qawn-5A* plays a major role in determining awn development. Moreover, the mapping interval of *Qawn-5A* IWB65661~IWB60644 was located in the region 697760461~698783143 on 5AL. Furthermore, the negative additive effect (Add) values of *Qawn-5A* and *Qawn-7B* (Table 2) suggested that these two QTLs are awn inhibitors from ZLWM. The positive Add value of *Qawn-1D* (Table 2) indicated that this QTL is offered by 4045 and may facilitate awn development or repress awn inhibitors.

### 2.3. Fine Mapping of Qawn-5A Locus

To obtain more SNP resources for marker developing and fine mapping, young spikes at Waddington 5.5 (W5.5) and Waddington 7.5 (W7.5) [29] stages and lemmas at the flowering stage were sampled and RNA-seq was performed. As a result, 221,544 SNPs were detected, of which 10,352 SNPs were located on chromosome 5A and 625 were located in the mapping region. A total of 128 SNPs that are homologous for both 4045 and ZLWM were selected for KASP marker development. To more precisely map the *Qawn-5A* locus, we extended the primary mapping interval and developed KASP markers at the positions of IWB65661 (3635 bp downstream), 684.75 kb downstream of IWB60644, and ~1.66 Mb downstream of IWB60644; namely Awn_SNP32, Awn_SNP21, and Awn_SNP27, respectively (Figure 3B, Appendix A). The three KASP markers were then employed to screen progenies of 4045 × ZLWM for recombinants among 661 field-grown F_2_ individuals. Five recombinants were identified between the three markers, which were divided into four types based on their genotypes and phenotypes (Figure 3B). The F_2_-a and F_2_-b types exhibited the same genotypes but different phenotypes, suggesting that *Qawn-5A* was downstream of Awn_SNP32 and upstream of Awn_SNP21. The F_2_-c type also proved that *Qawn-5A* was upstream of IWB60644, while the F_2_-d type recombinant between Awn_SNP21 and Awn_SNP27 further confirmed this conclusion (Figure 3B). Hence, the *Qawn-5A* locus was mapped to an interval between Awn_SNP32~Awn_SNP21, located at 697764096~699467893 on chromosome 5AL (Figure 3B, Appendix A).

To further narrow the mapping region, an F_2:3_ population containing 3505 individuals derived from heterozygous F_2_ plants was generated and screened with markers Awn_SNP32 and Awn_SNP21. Awn_SNP27 was also included to reconfirm the mapping result from the F_2_ population. As a result, 15 recombinants in five types were identified (Figure 3C). Furthermore, more KASP markers, including Awn_SNP66, Awn_SNP74, Awn_SNP87, and Awn_SNP101 were developed, to enrich the marker density within the mapping interval (Figure 3C). The F_2:3_-a and F_2:3_-b types showed 4045-type genotypes in the Awn_SNP32~Awn_SNP21 interval and awned phenotypes, reconfirming the previous mapping result. The other three types of recombinants—F_2:3_-c, F_2:3_-d, and F_2:3_-e—were between Awn_SNP32 and Awn_SNP66. These recombinants confirmed the conclusion that *Qawn-5A* was downstream of Awn_SNP32; nevertheless, we could not narrow the mapping region (Figure 3C).

To screen for more recombinants between Awn_SNP32 and Awn_SNP21, and to further narrow the mapping interval, 4852 F_3:4_ plants derived from heterozygous F_2:3_ were screened with Awn_SNP32 and Awn_SNP21. Meanwhile, two more KASP markers, Awn_SNP51 (447 bp downstream of Awn_SNP32) and Awn_SNP81 (445 bp upstream of Awn_SNP21), were designed to ensure that we obtained correct and robust genotypes (Figure 3D). According to the genotypes, a total of 28 recombinants in six types were identified (Figure 3D). Fortunately, two plants (F_3:4_-a type and F_3:4_-f type) that recombined between Awn_SNP66 and Awn_SNP74 were screened out, and *Qawn-5A* was mapped to an 809-kb region from 698510686~699320125 on 5AL (Figure 3D). This location is consistent with the previously reported *B1* locus [19].

### 2.4. Analysis of Candidate Genes in the Mapping Interval

According to the wheat reference genome of Chinese Spring (CS) Refseq_V1.1 (IWGSC 2018), five genes were annotated in the mapping region (Figure 3E). Taking into consideration that the awn segregation ratio in F_2_ was close to 3:1 (Figure 1D), the causal gene of the awnless phenotype is likely to be partially dominant. Consequently, the causal gene should express at the awn emergence and development stages. We then analyzed the transcriptional abundance of the five genes of young spikes at the W5.5 and W7.5 stages and lemmas at the flowering stage in 4045 and ZLWM. As a result, *TraesCS5A02G542700*, *TraesCS5A02G543000*, and *TraesCS5A02G543100* were rarely expressed in all of the samples, especially in the samples at W5.5 and W7.5 stages (Figure 4), implying that these genes are unlikely to be casual. *TraesCS5A02G542900* showed a higher expression level at all stages in ZLWM. However, a relatively high expression was detected for *TraesCS5A02G542800* at the W5.5 stage in ZLWM, while no expression was detected at the same stage in 4045. Its expression decreased at W7.5, compared to W5.5, in ZLWM but still had six times higher expression than that in 4045 (Figure 4). In lemmas at the flowering stage, *TraesCS5A02G542800* was slightly expressed in ZLWM, while no expression was detected in 4045 (Figure 4). To verify the expression patterns of the five genes, qRT-PCR was employed to detect the relative expression versus *TaActin* [30]. Consistent with the transcriptional expressions detected by RNA-seq, *TraesCS5A02G542700*, *TraesCS5A02G543000*, and *TraesCS5A02G543100* rarely expressed in all of the samples. *TraesCS5A02G542900* was detected only in young spike samples while its expression was much lower compared to *TraesCS5A02G542800*. *TraesCS5A02G542800* was highly expressed in young spikes of ZLWM but was barely detected in 4045 samples (Appendix A). These results strongly imply that *TraesCS5A02G542800* is the *B1* awn inhibitor at the *Qawn-5A* locus.

*TraesCS5A02G542800* encodes 121 amino acids in a single exon, which is annotated as a C_2_H_2_ zinc finger protein with two EAR motifs (Figure 5B). To gain deeper insight into the sequence information in ZLWM, we cloned and analyzed the sequence of both encoding and promoter regions of *TraesCS5A02G542800* in ZLWM and 4045 (Figure 5 and Appendix A). As expected, sequence analysis showed that compared to CS, which also performs an awnless phenotype, ZLWM has an identical encoding region, but a 25-bp deletion was detected 326 bp upstream of the start codon (Figure 5A and Appendix A). Nevertheless, two SNPs were detected in 4045: G148C caused an alteration from proline to alanine; however, another mutation, C322T, resulted in premature termination of translation as the codon CAA was converted into the stop codon TAA, removing the EAR2 motif from TraesCS5A02G542800 (Figure 5 and Appendix A). These mutations may lead to reduced function of the TraesCS5A02G542800 protein. Moreover, a 261-bp insertion was detected 613 bp upstream of the start codon in 4045 (Figure 5A and Appendix A), which might further decrease the expression of *TraesCS5A02G542800* in 4045.

### 2.5. Overexpression of TraesCS5A02G542800 in a “Double-Awn” Acceptor

To verify the hypothesis that *TraesCS5A02G542800* is the candidate gene at the *Qawn-5A*/*B1* locus, the 366-nt ORF driven by maize ubiquitin promoter was induced in a “double-awn” receptor wheat YM20 and 165 transgenic plants were obtained in the T_1_ generation. The awn length of the overexpression (OE) lines was then carefully measured. As expected, the awn length in transgenic lines was significantly shortened (Figure 6); that is, the lengths of the top awns, middle awns, bottom awns, and total awns of the spike were reduced by 63%, 60%, 56%, and 60%, respectively (see Figure 6C). This result demonstrated that *TraesCS5A02G542800* is the causal gene of the awn inhibition phenotype at the *Qawn-5A* locus. Notably, the OE lines retained the “double awn” phenotype although the length of both the lemma awns and glume awns were repressed (Figure 6A,B), indicating that *B1* limitedly suppressed the elongation of super-long lemma awns while had no impact on the emergence of awns.

To investigate the influence of different haplotypes of *B1* on the main agronomic traits, the phenotypes of length and width of flag leaf, spike length, tiller number, spikelet distance, plant height, spikelet number, peduncle length, and flag leaf sheath length of the 101 greenhouse-grown F_2_ plants were analyzed (Appendix A). According to the results, the ZLWM haplotype of the *B1* allele (B1_HapZLWM) significantly decreased the tiller number while increasing the spikelet number, peduncle length, and flag leaf sheath length (Appendix A). No significant differences were detected for flag leaf length and width, spike length spikelet distance, and plant height between b1_Hap4045 and B1_HapZLWM (Appendix A).

## 3. Discussion

Wheat awns are the long, solid, and slender extensions of the lemma, which play multiple roles, including in the dispersal of seeds, protection of the grain from predation by animals, and supplying carbohydrates to developing grains [1,6,7,8,13,31]. The morphological, developmental, and genetic features of the awns have been investigated by researchers for decades [6,19,20,21,22,32]. Cloning and characterizing the three inhibitors *B1*, *B2*, and *Hd* has become a hot topic in recent years. Before completing this work, the *B1* inhibitor had been finely mapped and characterized by four research groups [33,34,35,36]. DeWitt et al. identified a significant SNP that is very close to the *B1* QTL region through a genome-wide association study (GWAS). Further fine mapping placed the *B1* locus in a 127-kb region with biparental populations of hexaploid wheat. Combined with the analysis of mutant lines and gene expression, the function of the candidate gene *TraesCS5A02G542800* was characterized [33]. Huang et al. identified the *B1* locus through bulked segregant RNA sequencing of an F_2_ durum wheat population ST × GH. They characterized *B1* gene function by overexpression of *TraesCS5A02G542800* in durum wheat and mutant analysis of bread wheat and demonstrated that *B1* encodes a C_2_H_2_ zinc finger protein with EAR motifs playing a predominant role in awn inhibition [34]. Wang et al. fine-mapped *B1* to a 125-kb physical interval. They cloned and functionally characterized the candidate gene *TraesCS5A02G542800* and named it *Awn Length Inhibitor 1* (*ALI-1*). Importantly, they systematically analyzed the polymorphisms and haplotypes of *ALI-1*. According to their discovery, no variations were found in the coding region of *ALI-1* between awned and awnless wheat cultivars, while natural variations in the promoter account for awn inhibition of the *B1* locus. In general, the haplotype TAGA is responsible for the *B1* allele (proved by 186 out of 196 accessions) and hap-CGAG represents the *b1* allele (proved by 191 out of 196 accessions) [36]. Niu et al. also fine-mapped the *B1* locus and recognized a new haplotype that carries a mutation from C to T at 322 bp of the encoding region, which is the same as the corresponding genotype in 4045 (Figure 5 and Appendix A) [35]. In contrast with these previous studies, we genetically analyzed and functionally characterized the awn inhibitors of an awnless accession, ZLWM, and a “double-awn” accession, 4045 (Figure 1A). Primary mapping by an F_2_ population identified the *B1* (*Qawn-5A*) locus, but not for *B2* and *Hd* loci (Figure 2). According to the awn morphology, ZLWM presented hooked structures at the tip of the lemma; similarly, 4045 showed curved structures (Figure 1A), implying that both of these wheat lines may harbor the *Hd* locus, which produces curved, hooked awns [20]. This explains for the absence of the *Hd* locus in the mapping results. As for the lack of *B2* locus, one possible reason is that it does not exist in both of the parental lines, but further investigation is needed for confirmation. Fine mapping, analysis of gene structure, and expression and overexpression identified and characterized the candidate gene *TraesCS5A02G542800*, which is same as *B1* (Figure 3, Figure 4, Figure 5 and Figure 6). This result demonstrates that *B1* exists, in a widespread manner, naturally across the world.

Based on the haplotype analysis by Wang et al. [36], the genotype of ZLWM belongs to a hap-TAGA type with a 25-bp deletion (Figure 5A), which was detected in 53 out of 89 accessions. This result indicates that the ZLWM haplotype widely exists in cultivars all over the world. Nevertheless, the haplotypes of over 1000 wheat germplasms were explored in the previous studies where no variation was reported in the encoding region of *B1* [33,34,36]. In this study, two SNPs, G148C and C322T, were detected in 4045 (Figure 5 and Appendix A). G148C caused an alteration from proline to alanine, while C322T resulted in premature termination of translation, as the codon CAA was converted into the stop codon TAA and the EAR2 motif was removed from the B1 protein (Figure 5 and Appendix A). The same alterations have also been detected in Lankao 86 (LK86) [35]; however, the awn length of 4045 (10.7 cm; Figure 1A) was much longer than that of LK86 (about 5 cm) [35], which is the common awn length of most of the awned bread wheat cultivars, such as Fielder and AK58 (Appendix A). Besides, LK86 showed no glume awns [35]. Consequently, the lack of the EAR2 motif is probably not the reason for the “double-awns” in 4045. Except for the two SNPs, a 261-bp insertion was detected 613 bp upstream of the start codon. The expression of *B1* in young spikes of 4045 was similar to that in other awned cultivars or lines [33,34,35,36], suggesting that this insertion may not affect the expression level of *B1* in young spikes. Moreover, overexpression of *B1* in “double-awn” wheat YM20 only partly reduced the length of both the lemma awns and the glume awns, which is different from the previous studies [34,36], but did not abandon the formation of the glume awns, indicating that the increased expression of *B1* limitedly suppressed the elongation of the super-long lemma awn and *B1* did not determine the formation of glume awns. Furthermore, the very small proportion (<1%) of the “double-awn” phenotype in the F_2_ population suggests that more essential genes or loci, such as *Qawn-1D* and *Qawn-7B* (Table 2, Figure 2), are involved in the determination and development of the “double-awns”. 

In rice, a lack of awns is considered to be an important domestication trait facilitating grain harvest and storage [16]. In contrast, awns have not been domestically disused in wheat. In contrast, the advantages of awns in improving yield have been proposed [3,4,5,10,31,37,38,39]. Although many excellent works have characterized the predominant awn inhibitor *B1* [33,34,35,36], the mechanisms and pathways of how *B1* and other awn inhibitors regulate the development of awns are still far from clear. Hence, further work is necessary to illuminate the precise mechanisms.

## 4. Materials and Methods

### 4.1. Plant Materials and Growth Conditions

The awnless wheat ZLWM was provided by Professor Yizhe He (Northwest Agriculture and Forestry University, Yangling, China). The “double-awn” wheat line 4045 was conserved in our lab. A total of 661 4045 × ZLWM F_2_ plants for fine mapping were grown at the experimental farm of Northwest A&F University in Yangling, Shaanxi, China, which were watered and fertilized regularly. The other offspring of 4045 × ZLWM, including 101 F_2_ plants for primary mapping, 3505 F_2:3_ plants, and 4852 F_3:4_ plants, were grown in a greenhouse with a photoperiod of 14:10 light:dark hours and temperatures between 18–30 °C after vernalization for 45 days at 4 °C.

### 4.2. Genetic Map Construction and Primary Mapping

Genomic DNA of ZLWM, 4045, and 101 greenhouse-grown F_2_ plants was isolated using the CTAB method. The samples were genotyped with a Wheat90K SNP array containing 81,587 SNP markers by China Golden Marker (Beijing, China). Subsequently, SNPs with missing values in either of the two parental accessions (i.e., ZLWM or 4045) were deleted, and other successfully genotyped polymorphic SNPs were chosen for map construction. Then, 7636 polymorphic SNPs were binned at random using the Bin function of the IciMapping 4.1 software (https://www.isbreeding.net/software/?type=detail&id=18, accessed on 21 May 2016). Bin markers were then divided into 21 groups using anchor information. Next, the markers were ordered using the nnTwoOpt strategy and rippled with the default parameters of the Icimapping software. The QTLs of the awnless trait were identified using the Bip function of IciMapping with the inclusive composite interval mapping (ICIM) method.

### 4.3. KASP Marker Development and Genotyping

SNPs obtained from RNA-seq were used for KASP marker development. About 50 bp of the flanking sequences of the SNPs at both sides were uploaded to polymarker (http://www.polymarker.info/, accessed on 12 March 2017) for KASP primer designing. The designed markers were first checked for 42 F_2_ DNA, along with the parental, heterozygous, and empty controls for quality control. Markers with high genotyping efficiency were finally selected and used for further genotyping.

Genomic DNA of 661 F_2_ individuals, 3505 F_2:3_ plants, and 4852 F_3:4_ plants were isolated and the DNA samples were genotyped using the developed KASP markers for recombinant screening. All of the recombinants were resampled and regenotyped to confirm their genotypes.

### 4.4. RNA Sequencing and Gene Expression Analysis

Total RNA of lemmas at the flowering stage as well as young spikes at the W5.5 and W7.5 stages of 4045 and ZLWM was isolated using an RNAprep Pure Plant Kit (Tiangen Biotech, Beijing, China). RNA-seq was performed by Gene-Health Biotech (Shijiazhuang, China). GATK [40] was employed to identify the SNPs. Fragments per kilobase of transcript per million fragments mapped (FPKM) values were applied to measure the expression level of a gene or transcript by StringTie using the maximum flow algorithm. The expression of *TraesCS5A02G542700, TraesCS5A02G542800, TraesCS5A02G542900, TraesCS5A02G543000,* and *TraesCS5A02G543100* were verified via qRT-PCR. The RNA samples were reverse-transcribed using a StarScript II First-strand cDNA Synthesis Kit (GeneStar, Beijing, China). The primers used were listed in Appendix A and the primers of *TaActin*, *TraesCS5A02G542700* and *TraesCS5A02G542800* were previously described [33,41]. qRT-PCR was performed with ChamQ SYBR qPCR Master Mix (Vazyme, Nanjing, China) in a QuantStudio 3 Real-Time PCR System (Applied Biosystems). Gene expression was normalized to expression relative to the endogenous control *TaActin* with 2^(TaActin CT − Target CT)^.

### 4.5. Gene Cloning and Structure Analysis of TraesCS5A02G542800

The promoter and encoding region of *TraesCS5A02G542800* were amplified from the genomic DNA samples of 4045 and ZLWM using KOD DNA polymerase (TOYOBO, Osaka, Japan) with the primers listed in Appendix A. The obtained DNA fragments were subsequently cloned into a pMD18-T entry vector (TAKARA) and sequenced by the Sanger sequencing strategy.

### 4.6. Generation of Overexpression Transgenic Wheat 

The overexpression construct of *TraesCS5A02G542800* was generated by amplifying the coding DNA sequence (CDS) fragment (366 bp) with the primers listed in Appendix A and by cloning the fragment into the pGA3426 plasmid [42] using a NovoRec^®^ plus One step PCR Cloning Kit (Novoprotein, Shanghai, China). The plasmid construct was transformed into *Agrobacterium tumefaciens* EHA105. Wheat plants (cv. YM20) were transformed using the shoot apical meristem method [43].

## Figures and Tables

**Figure 1 plants-10-02588-f001:**
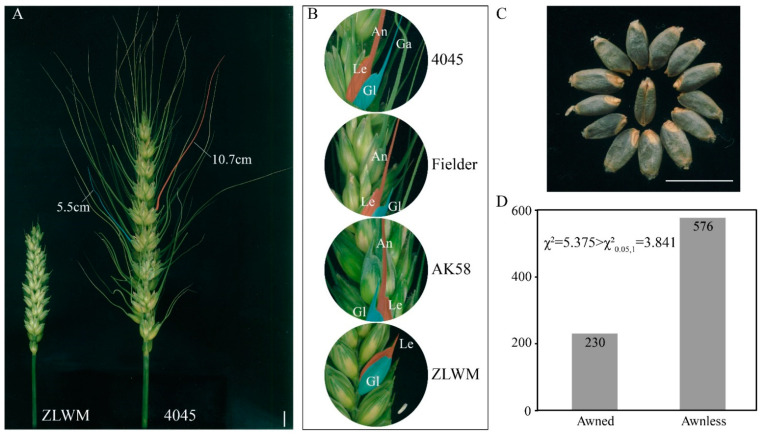
(**A**) Spike morphology of awnless wheat ZLWM and long-awn wheat 4045. The awn indicted by a red shadow is an example of middle-spikelet awns, while the awn indicted by a blue shadow is an example of glume awns. (**B**) Morphological features of lemma awn and glume awn in different wheat cultivars. Le, lemma; Gl, glume; An, lemma awn; Ga, glume awn. (**C**) Blue-colored grains of ZLWM. (**D**) Segregation of 4045 × ZLWM F_2_ population. Bars = 1 cm.

**Figure 2 plants-10-02588-f002:**
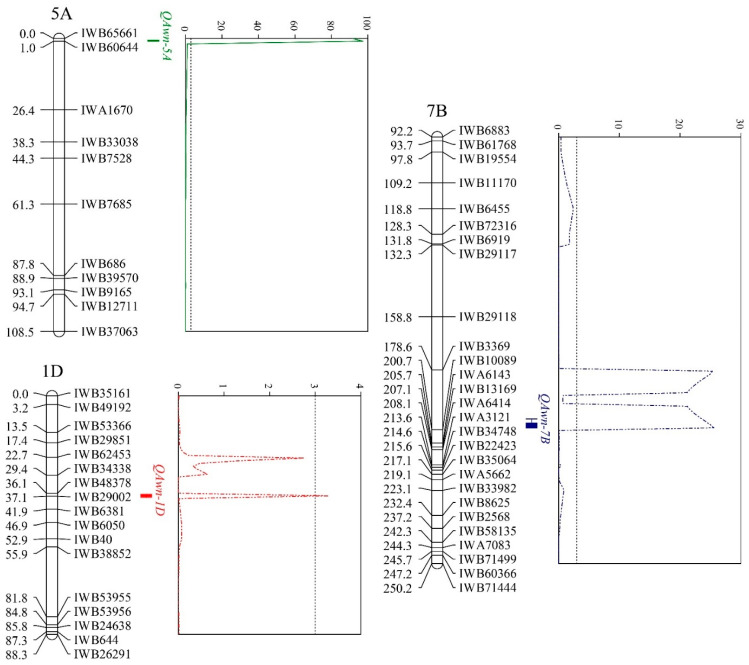
QTL mapping results of awnless trait. QTL plots are presented next to the linkage maps. X-axis = positions of the linkage map; Y-axis = LOD values.

**Figure 3 plants-10-02588-f003:**
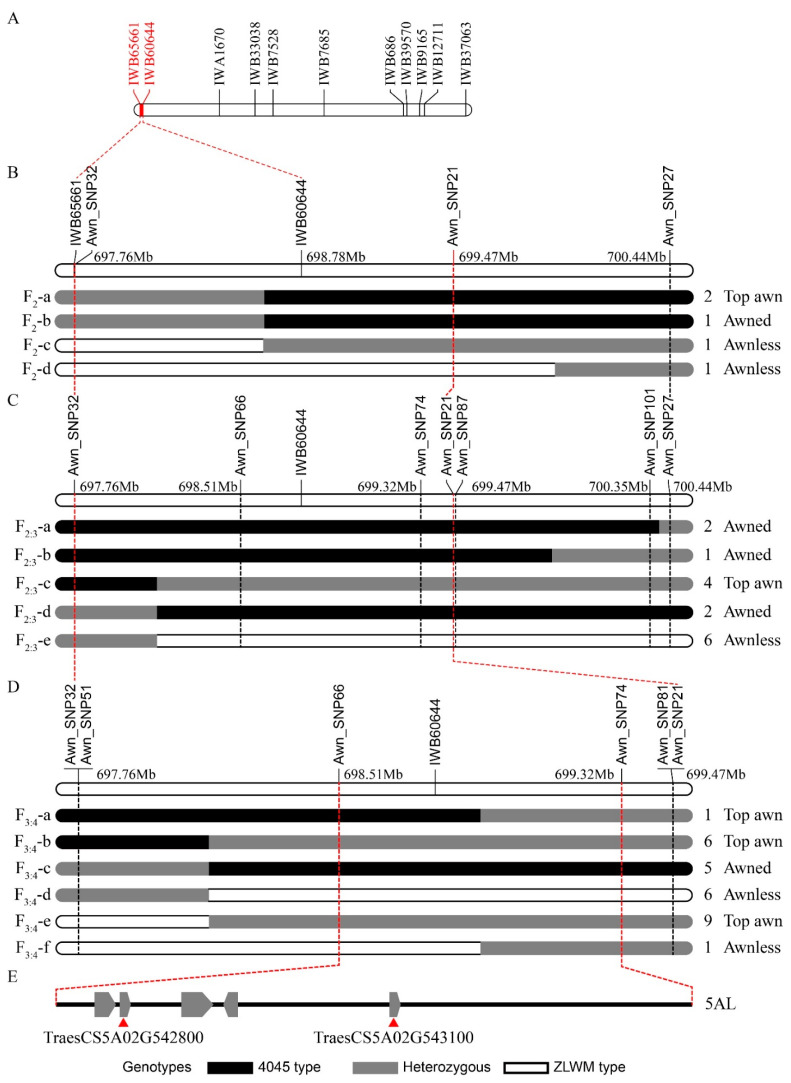
Fine mapping of *Qawn-5A* locus: (**A**) Genetic linkage map of 5A chromosome. (**B–D**) Fine mapping details from F_2_, F_2:3_, and F_3:4_ populations, respectively, of 4045 × ZLWM. The numbers of recombinants and the phenotype of each genotype are indicated on the right. Recombinant types are indicated on the left. (**E**) Physical map of the mapping region on chromosome 5AL based on Refseq_V1.1.

**Figure 4 plants-10-02588-f004:**
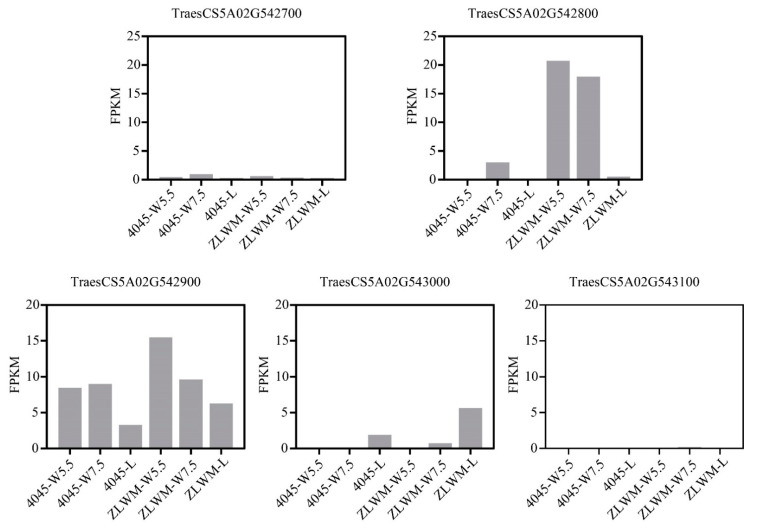
The transcriptional abundance of the five genes detected by RNA-seq in young spikes and mature lemmas of ZLWM and 4045.

**Figure 5 plants-10-02588-f005:**
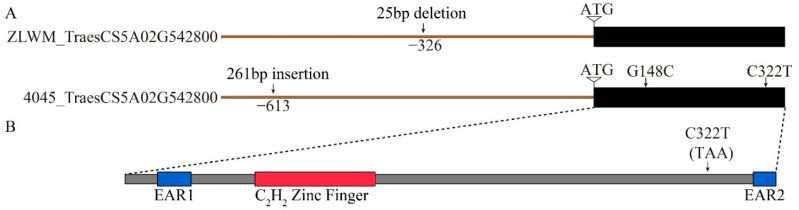
Structures of candidate gene *TraesCS5A02G542800*: (**A**) Gene structure of *TraesCS5A02G542800* in ZLWM and 4045. Mutations in encoding or promoter regions, compared to CS genome (Refseq_V1.1), are indicated. (**B**) Protein structure of *TraesCS5A02G542800*.

**Figure 6 plants-10-02588-f006:**
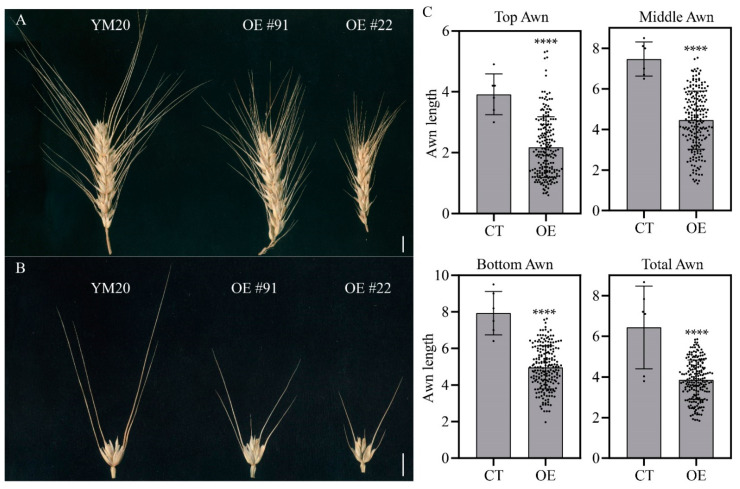
Phenotypes of the overexpression lines of *TraesCS5A02G542800*: (**A**) Spikes of the OE lines and their accepter YM20. Bar = 1 cm. (**B**) Spikelets isolated from spikes of the corresponding OE lines in (**A**). Bar = 1 cm. (**C**) Awn length of OE lines. Awn lengths at the top, middle, and bottom parts of the spike were measured. *n* = 6 for the control (cv. YM20), *n* ≥ 165 for OE lines. Each spot indicates a value of awn length. **** indicates *p* < 0.0001 by Student’s *t*-test.

**Table 1 plants-10-02588-t001:** Information regarding the high-density genetic map constructed using a 4045 × ZLWM F_2_ population.

Chromosome	No. of Bins	No. of SNPs	No. of Linkage Groups *	Length (cm)	Density	Max Interval (cm)
1A	172	289	4	829.25	4.82	27.79
1B	225	389	7	1105.54	4.91	26.43
1D	118	314	8	678.66	5.75	27.12
2A	212	561	6	1148.53	5.42	29.69
2B	242	626	8	1167.14	4.82	28.88
2D	99	365	13	415.6	4.20	25.29
3A	198	416	9	977.92	4.94	26.03
3B	318	607	3	1443.05	4.54	28.6
3D	42	79	8	390.64	9.30	35.62
4A	148	310	9	991.85	6.70	29.16
4B	130	233	8	688.47	5.30	27.89
4D	25	89	3	124.95	5.00	19.05
5A	164	372	7	977.08	5.96	29.21
5B	370	875	5	1332.7	3.60	28.47
5D	21	32	5	187.41	8.92	25.21
6A	164	373	5	1035.51	6.31	28.46
6B	232	505	3	1010.31	4.35	29.6
6D	18	33	3	87.64	4.87	26.57
7A	207	507	8	1056.22	5.10	27.8
7B	171	401	6	737.07	4.31	28.87
7D	11	13	3	53.65	4.88	24.54
A genome	1265	2828	48	7016.36	5.55	29.69
B genome	1688	3636	40	7484.28	4.43	29.6
D genome	334	925	43	1938.55	5.80	35.62
Whole genome	3287	7389	131	16,439.19	5.00	35.62

* The linkage map of each chromosome was split into different numbers of linkage groups.

**Table 2 plants-10-02588-t002:** QTLs identified with respect to awn trait.

QTLs	Chromosome	Position	Left Marker	Right Marker	LOD	PVE (%)	Add
*Qawn-1D*	1D	37	IWB48378	IWB29002	3.2713	0.6319	0.0699
*Qawn-5A*	5A	1	IWB65661	IWB60644	97.3171	91.2205	−0.5019
*Qawn-7B*	7B	200	IWB3369	IWB10089	25.5369	2.7552	−0.0106

## Data Availability

The RNA-seq reads have been deposited into GenBank under bioProject PRJNA771461.

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
