# Peer review of "Mapping and Characterization of QTLs for Awn Morphology Using Crosses between “Double-Awn” Wheat 4045 and Awnless Wheat Zhiluowumai"

_plants, 2021, doi:10.3390/plants10122588_

Round 1

Reviewer 1 Report

The awn development in plants/crops has been studied in great detail, but not too many confident genes have been identifed in terms of genetic and functional findings. This manuscript revealed the morphologic and genetic characterizations of wheat awn using “double awn” and awnless variaties. The authors identified that three QTLs may associate with awn phenotype. They also provided the functional analysis of candidate genes using RNA-seq and transgenic identification. Overall, the work presented is very interesting, the approach taken is valid and the data is of good quality. Where appropriate, statistics, genetics have been applied. The references cited credit previous work appropriately. The manuscript is well-written and reasonably accessible to a reader who works on plant/crop development and breeding research. Detaied comments are listed below:

  • In introduction, Line 83-84, the authors should describe the details of candidate gene, TraesCS5A02G542800, such as the encoding protein and variation allele.
  • For reseult section of 2.4, the expression of 5 genes annotated in the mapping region only showed from RNA-seq database (FPKM), the authors should provide the validation analysis using qRT-PCR.
  • For W2.5, the expression data of TraesCS5A02G542800 in OE transgenic plants is missing. The overexpressed candidate gene in selected transgenic lines is required, in comparsion with WT.
  • Any figures of awn phenotype in OE transgenic plants could be shown ?

Author Response

Response to Reviewer 1 Comments

Point 1: In introduction, Line 83-84, the authors should describe the details of candidate gene, TraesCS5A02G542800, such as the encoding protein and variation allele. 

Response 1: Thank you for the suggestion. We have modified the corresponding text accordingly. See line 91-100 of the revised manuscript.

Point 2: For reseult section of 2.4, the expression of 5 genes annotated in the mapping region only showed from RNA-seq database (FPKM), the authors should provide the validation analysis using qRT-PCR.

Response 2: We performed qRT-PCR and verified the transcriptional abundance of the five genes, see Figure S3 in the revised supplementary material file. And the corresponding contents in the Results, Materials and Methods, and Discussions were revised as well.

Point 3: For W2.5, the expression data of TraesCS5A02G542800 in OE transgenic plants is missing. The overexpressed candidate gene in selected transgenic lines is required, in comparison with WT. Any figures of awn phenotype in OE transgenic plants could be shown.

Response 3: Thank you for the insight and great suggestions. We obtained the spikes of OE transgenic lines from our collaborator. The spikes and awns of OE lines and WT (cv. YM20) were imaged and presented in Figure 6A. We have planted the seeds of OE lines; however, it takes months for them to grow to the awn development stages. Thus, we are not able to detect the expressions of TraesCS5A02G542800 for the OE lines at present.

Moreover, we updated the Materials and Methods section as it was not satisfactory in the old version.

Reviewer 2 Report

Dear Authors,

a comprehensibly written paper which clearly demonstrates your study and results. I do have some comments/questions:

  • in line 139 of the pdf you state a LOD score of 97.32 and also further down the explained variance has the same value
  • in table 1 no of groups is nowhere explained and not quite clear to me
  • how many SNPs did you initially detect from your RNA seq? Might be worth mentioning if different from the SNPs which you finally used for fine-mapping
  • line 269 (pdf): I am sure that needs to be 127 kb and not bp
  • I also did some corrections of language in the attached word file, but please show the paper to a native speaker, it deserves some further language editing

Author Response

Response to Reviewer 2 Comments

Point 1: In line 139 of the pdf you state a LOD score of 97.32 and also further down the explained variance has the same value.

Response 1: Thank you for picking out the written mistake. We have corrected it.

Point 2: In table 1 no of groups is nowhere explained and not quite clear to me.

Response 2: We have changed the heading “No. of groups” to “No. of linkage groups” and added a note of it below table 1 to make it clear. When construct the genetic linkage map, markers from a chromosome may be grouped into different linkage groups due to the existence of low-recombination or non-recombination regions.

Point 3: For how many SNPs did you initially detect from your RNA seq? Might be worth mentioning if different from the SNPs which you finally used for fine-mapping

Response 3: Thank you for the suggestion. Detailed information of SNPs has been added to the text.

Point 4: Line 269 (pdf): I am sure that needs to be 127 kb and not bp

Response 4: Thanks for picking out the error. We have corrected it.

Point 5: I also did some corrections of language in the attached word file, but please show the paper to a native speaker, it deserves some further language editing

Response 5: Thanks a lot for the revisions. We modified the manuscript accordingly. And the paper has undergone English language editing by the MDPI English editing service.

Reviewer 3 Report

In addition to information on the authors' important results, the abstract should include information about any other interesting discoveries.

Despite the fact that the writers present a thorough evaluation of the literature in the part of the Introduction, they should strengthen and rebuild the section of the Introduction. They should be specific about the scope of the paper they are writing.

A common theme is the repeating of information that may have been overlooked the first time around.

The abstract and conclusion need to be changed completely. Only one phrase in the abstract portion is devoted to the outcomes, while the remainder of the section is devoted to the problem statement and its solutions.

The conclusion is just a reiteration of the issue statement, which does not make any sense in this context.

The statistical analysis should contain information on data distribution and the pre-requisites, as well as other relevant information.

Examine the ligands in the illustrations; they have been scrawled on the page at random.

When discussing notable and related works, there should be a greater amount of material and references included.

For the discussion part, I recommend that the style be simplified and that the material be made more legible by removing lengthy phrases.

Author Response

Response to Reviewer 3 Comments

Point 1: In addition to information on the authors' important results, the abstract should include information about any other interesting discoveries.

Response 1: We re-wrote the abstract according to your comments. A summary about discoveries was added to the revised abstract.

Point 2: Despite the fact that the writers present a thorough evaluation of the literature in the part of the Introduction, they should strengthen and rebuild the section of the Introduction. They should be specific about the scope of the paper they are writing.

Response 2: Thank you the great suggestions about writing skills. The introduction has been re-organized and the paper was English-edited to improve the descriptions. Now, in general, the introduction contains three parts. The first part highlighted the importance of the awns for grasses and the crops; the second part summarized discoveries about the identified genes involved in awn development in grasses especially the awn inhibitors in wheat. The last part introduced the materials, and the main discoveries and contributions of this study.

Point 3: A common theme is the repeating of information that may have been overlooked the first time around.

Response 3: The repeating information was removed in the revised version.

Point 4: The abstract and conclusion need to be changed completely. Only one phrase in the abstract portion is devoted to the outcomes, while the remainder of the section is devoted to the problem statement and its solutions.

Response 4: Thank you for the suggestions. The abstract has been completely updated according to your comments.

Point 5: The conclusion is just a reiteration of the issue statement, which does not make any sense in this context.

Response 5: As mentioned in the manuscript template of the journal, the conclusion section is not mandatory but can be added to the manuscript if the discussion is unusually long or complex. Since the discussion section of this paper was not that long or complex, we removed the conclusion part.

Point 6: The statistical analysis should contain information on data distribution and the pre-requisites, as well as other relevant information.

Response 6: Yes. We have corrected the legends of Figure 6 and Figure S5. More relevant information was added.

Point 7: Examine the ligands in the illustrations; they have been scrawled on the page at random.

Response 7: Thank you. We have examined and corrected them.

Point 8: When discussing notable and related works, there should be a greater amount of material and references included.

Response 8: Thank you for the advice. The discussion has been updated and more detailed descriptions and deeper discussions were added.

Point 9: For the discussion part, I recommend that the style be simplified and that the material be made more legible by removing lengthy phrases.

Response 9: Thank you for the suggestions, which helped us to improve the manuscript. We modified the discussion part according to the comments. Besides, the paper has undergone English language editing by the MDPI English editing service and was greatly improved.

Round 2

Reviewer 3 Report

The authors have answered most of my concerns; therefore manuscript can be for publication after a careful editorial check.

Author Response

Point 1: The authors have answered most of my concerns; therefore manuscript can be for publication after a careful editorial check.

Response 1: Thank you! The manuscript has been carefully re-edited.